# Mechanochemical Characterisation of Calcined Impure Kaolinitic Clay as a Composite Binder in Cementitious Mortars

**Kwabena Boakye [1], Morteza Khorami [1,\*], Messaoud Saidani [1], Eshmaiel Ganjian [2], Andrew Dunster [3], Ahmad Ehsani [4] and Mark Tyrer [5]**

1   School of Energy, Construction & Environment, Faculty of Engineering & Computing, Institute for Future Transport and Cities, Coventry University, Coventry CV1 5FB, UK; boakyek4@uni.coventry.ac.uk (K.B.); cbx086@coventry.ac.uk (M.S.)
2   Concrete Corrosion Tech Ltd., Birmingham B17 0JN, UK; eganjian@yahoo.co.uk
3   Building Research Establishment (BRE), Hertfordshire WD25 9XX, UK; andrew.dunster@bregroup.com
4   Indo-UK Centre for Environmental Research and Innovation, University of Greenwich, Chatham ME4 4TB, UK; a.ehsani@greenwich.ac.uk
5   Institute of Advanced Study (Collegium Basilea), 4053 Basel, Switzerland; m.tyrer@mtyrer.net
\*   Correspondence: aa8186@coventry.ac.uk

**Abstract:** The availability of some supplementary cementitious materials, especially fly ash, is of imminent concern in Europe due to the projected closure of several coal-fired power generation plants. Pure kaolinitic clays, which arguably have the potential to replace fly ash, are also scarce and expensive due to their use in other industrial applications. This paper examines the potential utilisation of low-grade kaolinitic clays for construction purposes. The clay sample was heat-treated at a temperature of 800 °C and evenly blended with Portland cement in substitutions of 10–30% by weight. The physical, chemical, mineralogical and mechanical characteristics of the blended calcined clay cement were determined. The Frattini test proved the pozzolanic potential of the calcined impure clay, as a plot of its CaO and OH⁻ was found below the lime solubility curve. The 28 days compressive strengths trailed the reference cement by 5.1%, 12.3% and 21.7%, respectively, at all replacement levels. The optimum replacement level between the three blends was found to be 20 wt.%.

**Keywords:** calcined clay; supplementary cementitious materials; low-grade kaolinitic clay; compressive strength; calcination

## 1. Introduction

The need for infrastructure development due to global population growth has resulted in a high demand for cement and cementitious products. Global Portland cement production was estimated to be 4.1 billion tonnes in 2019 [1] and has been projected to increase to about 4.5 billion tonnes by 2050 [2]. However, the production of Portland cement is known to contribute to the release of harmful gases into the environment, contributing to the depletion of the ozone layer and global warming [3,4]. The production of a tonne of Portland cement is estimated to generate a similar amount of carbon dioxide into the environment [3,5–8]. This constitutes about 4–8% of all global emissions [8,9]. To find solutions to this environmental concern, several studies have been conducted over the years to find alternative materials, which can be used to replace or supplement Portland cement in concrete. Partially substituting Portland cement with supplementary cementitious materials (SCMs) has proven to be an effective method in reducing Portland cement production and achieving sustainable development as well, by reducing the embodied energy and carbon footprint of concrete. The utilisation of supplementary cementitious materials in concrete has several benefits, including a reduced heat of hydration, improved workability, strength, durability and, ultimately, increased sustainability [9]. Studies have proven that

SCMs have an influence on cement hydration kinetics (by an initial filler effect), hydrated phases leading to phase assemblage and C-S-H compositions. During the hydration of cementitious systems, SCMs play a major role by reducing the calcium-to-silicon ratio of C-S-H while improving the mean silica chain length [10,11]. Their presence in concrete also significantly alters the microstructure by refining the pore structure, making it more dense [12]. SCMs are essential materials in cementitious systems for controlling the physical and chemical properties and, consequently, their mechanical and durability performance. Their physical and chemical make-ups permit their usage as a part replacement for Portland cement in concrete. They also alleviate the disposal difficulties of several industrial by-products including ground granulated blast-furnace slag (GGBS), fly ash, silica fume, etc. [13]. One SCM, which has become widely known in recent times, is calcined clays.

Calcination is the burning of clay at a particular temperature to allow its minerals to undergo dehydration and structural transformation [13,14]. This dehydration decreases the bond between the atoms in the octahedral sheet, thereby improving its reactivity [15]. This structural transformation is influenced by the calcination temperature, heating rate and time, atmospheric conditions and cooling process [16–18]. Critical attention should be given to the calcination temperature since an insufficient temperature may hinder dehydroxylation and excessive temperature could cause recrystallization [19–21].

Calcined clay, when used as an SCM in mortar and concrete, has several advantages such as improved density, low heat of hydration, improved mechanical properties and appreciable durability performance [22–26]. High-grade kaolinitic clays, often effective as SCM, are scarce and can be found in specific places and are also costly because of their use in other industrial processes [27]. However, most clay materials contain kaolinite (mostly in smaller quantities), smectite, illite and other impurities. Ridding clay of such impurities may be costly, energy-intensive and time-consuming. There is therefore the need to extensively consider the utilisation of impure calcined kaolinitic clays as a cement substitute in concrete applications.

Several researchers [14,15,28–33] have explored the potential of utilising impure clays such as SCM for Portland cement substitution. The effectiveness of clay used as SCM is generally influenced by its kaolinite content. There has not been any clear distinction in what constitutes a high or low grade in terms of its kaolinitic content. However, most researchers consider a kaolinite content of less than 40% to be low-grade. Zhou et al. [15] studied the use of calcined excavated waste clays (calcined between 600 °C–1000 °C) as a replacement in concrete and reported an improved compressive strength after curing for 28 days. Some researchers [29,30], on the other hand, have shown decreasing or similar compressive results as for the reference cement.

This research explores the possibility of using a particular impure clay, having a kaolinitic content of 17%, as supplementary cementitious material for mortar applications. The clay was heat-treated at 800 °C and used to replace Portland cement in substitutions of 10–30% by weight. The physical, chemical, mineralogical and mechanical characteristics were determined. The outcome of this study would contribute to forming a roadmap for the use of impure clays in cementitious systems, especially in areas where pure kaolinitic clays are scarce.

## 2. Materials and Methods

### 2.1. Materials

Portland cement CEM I 52.5 N conforming to BS EN 197-1 and manufactured by Hanson (Heidelberg Cement Group) was the main binder used for all the paste and mortar mixes. The Portland cement had a specific surface area of 410 m$^2$/kg. The chemical composition, as determined by X-ray fluorescence (XRF), is displayed in Table 1. Calcined clay was produced from naturally occurring clay, obtained from H.G Matthews Ltd., a brick manufacturing company in Bellingdon, England. The chemical composition of the clay is also presented in Table 1.

**Table 1.** XRF of raw materials.

| Material | Chemical Composition, % | | | | | | | | | |
|---|---|---|---|---|---|---|---|---|---|---|
| | $SiO_2$ | $Al_2O_3$ | $Fe_2O_3$ | MgO | CaO | $Na_2O$ | $K_2O$ | $SO_3$ | Cl | LOI |
| Raw clay | 59.95 | 18.2 | 6.51 | 1.3 | 0.14 | 1.6 | 1.29 | 0.07 | 0.01 | 10.9 |
| CEM-1 | 21.0 | 4.4 | 2.7 | 1.6 | 64.7 | 0.6 | 1.99 | 2.27 | 0.01 | 0.73 |

*2.2. Methods*

The clay sample was firstly oven-dried to remove moisture and crushed into smaller sizes using a hammer mill. It was then calcined for 2 h in a furnace at a temperature of 800 °C at a heating rate of 10 °C/min. This calcination temperature was selected based on previous studies of other researchers [34]. Using a laboratory-type ball mill, the calcined clay was milled to obtain cement fineness. The BET fineness of the calcined clay was found to be 21.3 $m^2$/g, whereas its specific gravity was 2.6. A Panalytical Axios mAX WDXRF spectrometer, manufactured by Malvern Panalytical Company Ltd., Worcestershire, UK, was used to determine the chemical composition of the materials. X-ray diffraction (XRD) analyses of the powders were also carried out using a 3rd generation Malvern Panalytical Empyrean XRD Diffractometer, manufactured by Malvern Panalytical Company Ltd., Worcestershire, UK. Thermal analysis (TG/DSC) was also conducted with the Perkin Elmer DSC 7 analyser, manufactured by PerkinElmer Inc., Waltham, MA, USA. The Malvern Mastersizer 2000, manufactured by Malvern Panalytical Company Ltd., Worcestershire, UK, was also used to determine the particle size distribution (PSD) of the raw materials.

A Frattini test was conducted with reference to BS EN 196-5:11. 20 g of the blended cement (containing a blend of 20% calcined clay and 80% Portland cement) was mixed with 100 mL deionised water in plastic containers, sealed tightly and kept for 7 and 28 days in a control environment with a temperature of 40 °C. The mixture was vacuum-filtered and analysed by titration using methyl orange as the indicator to determine the amount of $Ca^{2+}$ and $OH^-$.

CEM-I cement was partly substituted with the calcined clay in proportions of 10–30% by weight to form blended cement. $50 \times 50 \times 50$ mm mortar cubes were prepared following methods prescribed by BS 196-1:2016, using a binder/sand ratio of 1:3 and water/cement ratio of 0.5. The mortar cubes were cured under water and their respective compressive strengths determined after 3, 7 and 28 days. The setting times and water demand were also determined by the Vicat method, as described in BS 196-3:2016.

**3. Results and Discussion**

Apart from the chemical composition and structural make-up of hydrated aluminosilicates, the reactivity of pozzolans can be significantly affected by how the particle sizes are distributed [35]. The PSD of the clay, calcined clay and CEM-1 cement are presented in Figure 1. The clay was observed to have coarser particles than the reference cement. It was used in its natural form without separating the clayey part from the rest of its constituents (gravel, silt and sand) by using any physical or chemical method. This accounts for the coarse nature of the particle size distribution of the clay. The d10 and d90 values of the calcined clay were found to be smaller when compared to the natural clay.

Figure 2 demonstrates the TGA/DSC analysis of the clay. An appreciable mass loss of about 1.71% is seen between 50–100 °C, which is usually the evaporation of free water [2,36]. Further removal of water occurs between 100 °C and 400 °C and can be associated with the pre-hydroxylation process due to the reorganization of the octahedral layer [31]. As the temperature is increased, there is a mass loss of 2.31%, and a wide exothermic peak shows up between 458 °C and 531 °C. This typically corresponds to the structural transformation of kaolinite to form metakaolinite [37]. The kaolinite content can be estimated as 17%, calculated for the mass loss and molecular weights of water and kaolinite.

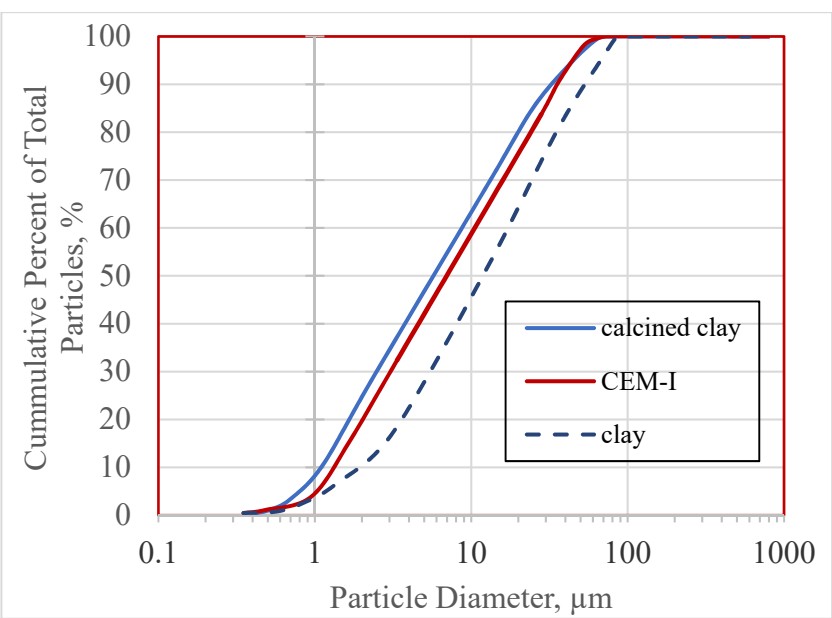

**Figure 1.** PSD of raw clay, calcined clay and CEM-I.

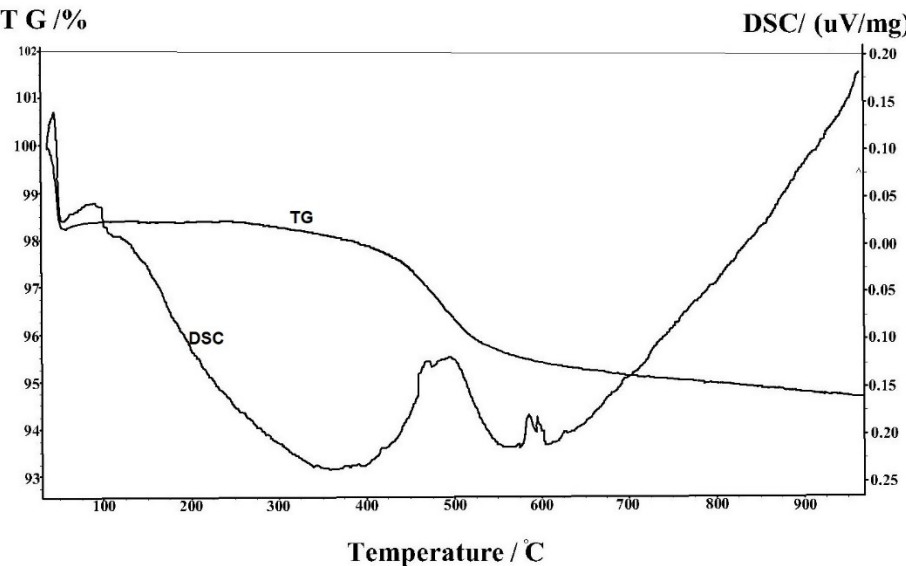

**Figure 2.** TG/DSC analysis of raw clay.

The scanning electron microscopy of the raw and calcined clay are shown in Figures 3 and 4, respectively. Typically, clay mineral particles have an irregular pseudo-laminar structure with a flat surface [15]. Heat treatment at 800 °C, however, broke up the clay particle structure and revealed a thick flaky-like structure. This flaky effect, after calcination, could be due to damage of the kaolinite plate, particle agglomeration, irregular shapes and undefined edges of the clay [31]. There also appears to be gaps found in between microscopic layers of the calcined clay. These interlayer gaps could be responsible for low workability in calcined clay mortar and concrete by entrapping water molecules [28]. This consequently reduces the water molecules available for interaction between the blended cement and water to form a workable paste.

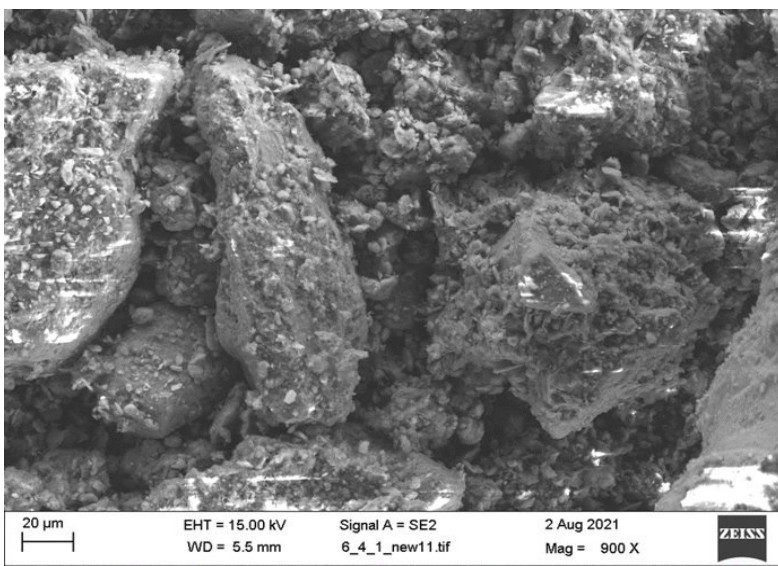

**Figure 3.** SEM image of raw clay.

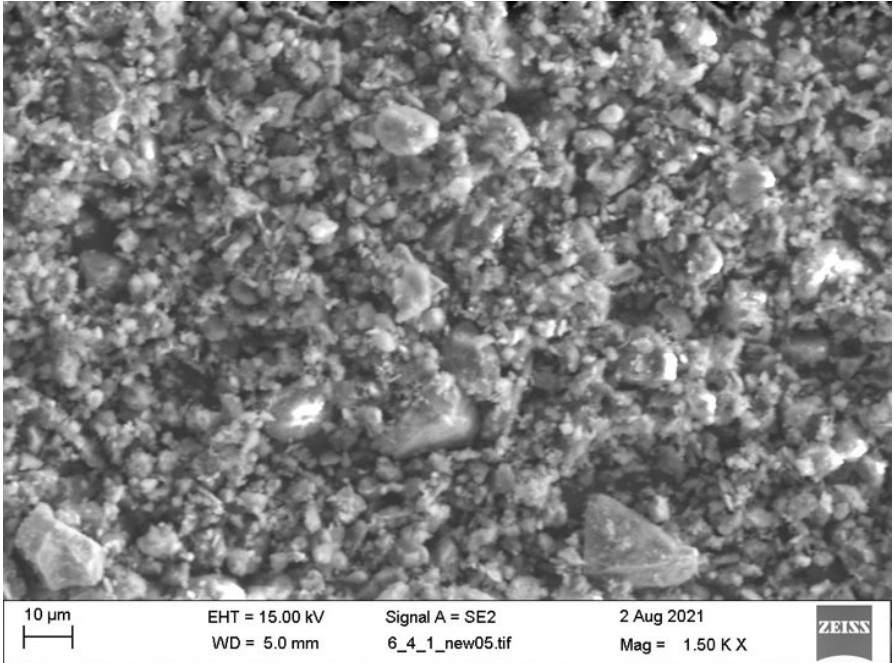

**Figure 4.** SEM image of clay calcined at 800 °C.

The X-ray diffraction analysis of the clay is displayed in Figure 5. It largely consists of quartz, kaolinite, illite, montmorillonite and other associated minerals. The presence of illite and montmorillonite can be associated with $K_2O$ and $Na_2O$/MgO [29], as seen in the XRF data (Table 1). Calcining at 800 °C led to the transformation of kaolinite from its crystalline phase to an amorphous phase, causing the kaolinite peak to disappear, signifyinging dehydroxylation. This phenomenon is in line with observations by other researchers [15,28,34]. The transformation of kaolinite to metakaolinite could be a result of the loss of the surface –OH group and rearrangement of Si and Al atoms, leading to the reduction of octahedral Al and the appearance of penta- and tetra-coordinate Al [38].

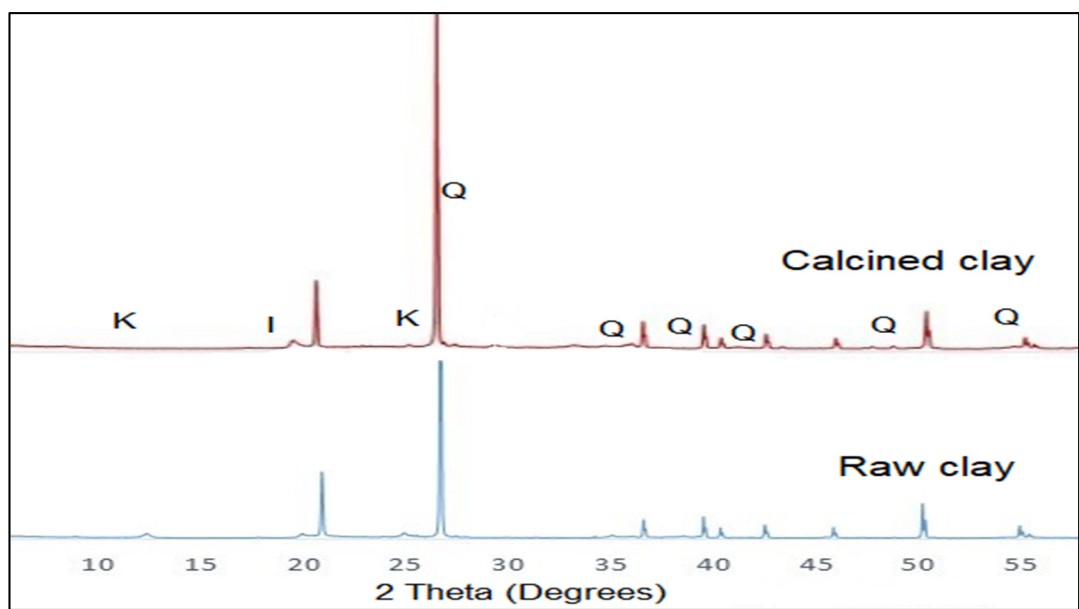

**Figure 5.** XRD of raw and calcined clay (*Q-Quartz; K-Kaolinite; I-Illite*).

The Frattini test, showing the reactivity of the heat-treated clay, is presented in Figure 6. It shows a plot of CaO against $OH^-$ [39]. An inert material (sand) is also included and used as a reference material. The clay calcined at 800 °C and blended with CEM-I cement showed a slight pozzolanic activity at seven days because a plot of their values is found just under the solubility curve. There appears to be an improved reactivity at 28 days, since the plot for calcined clay is located further below the lime solubility curve. This shows that the $Ca(OH)_2$ emerging from the reaction between cement and the pozzolan has been used up, indicating the pozzolanic potential of the material. Sand, however, is seen in the oversaturated region above the lime solubility curve, indicating no pozzolanic reaction.

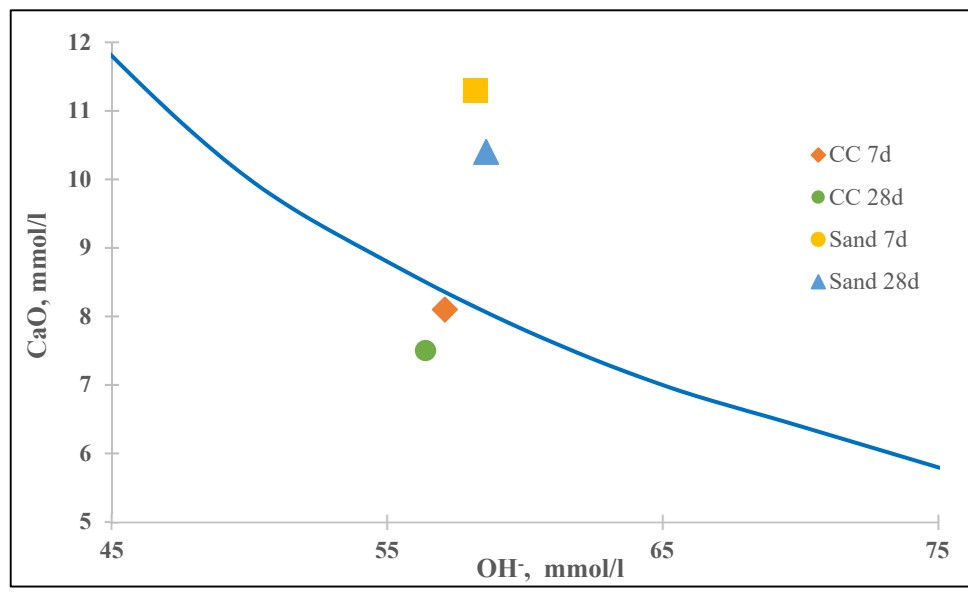

**Figure 6.** Frattini test showing the reactivity of the calcined clay.

A plot of the setting time and water demand is shown in Figures 7 and 8, respectively. CEM-I cement recorded initial and final setting times of 158 min and 245 min, respectively. This, however, consistently increased, as the calcined clay replacement increased from 10 wt.% to 30 wt.% in the mortar mix. Similarly, there was a progressive increase in the

water demand, as calcined clay content increased. The incorporation of calcined clay in mortar and concrete is generally known to affect the water demand and, consequently, the setting time and workability. This could be attributed to its high reactivity, specific surface area and amorphous structure [33].

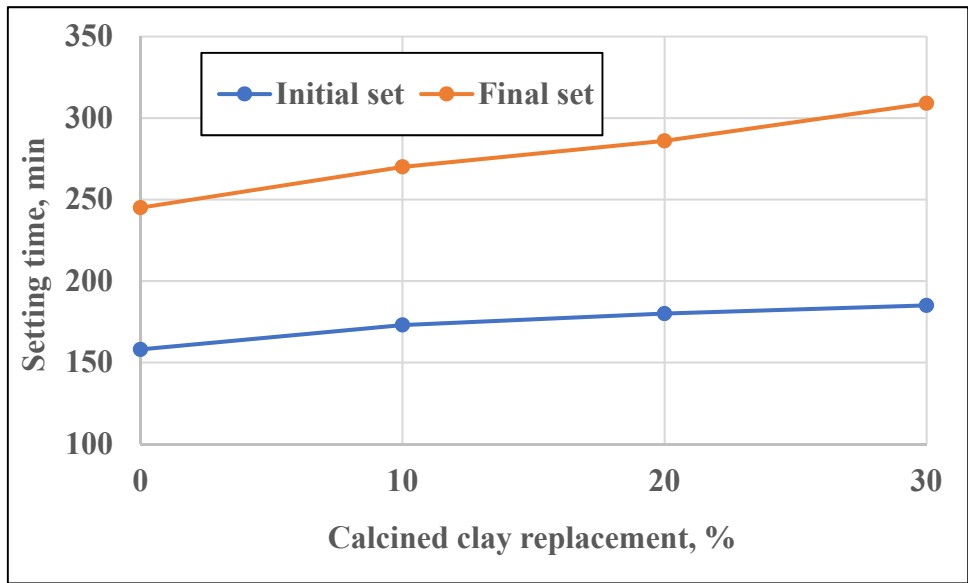

**Figure 7.** Setting times of calcined clay blended cements with varying percentage replacements.

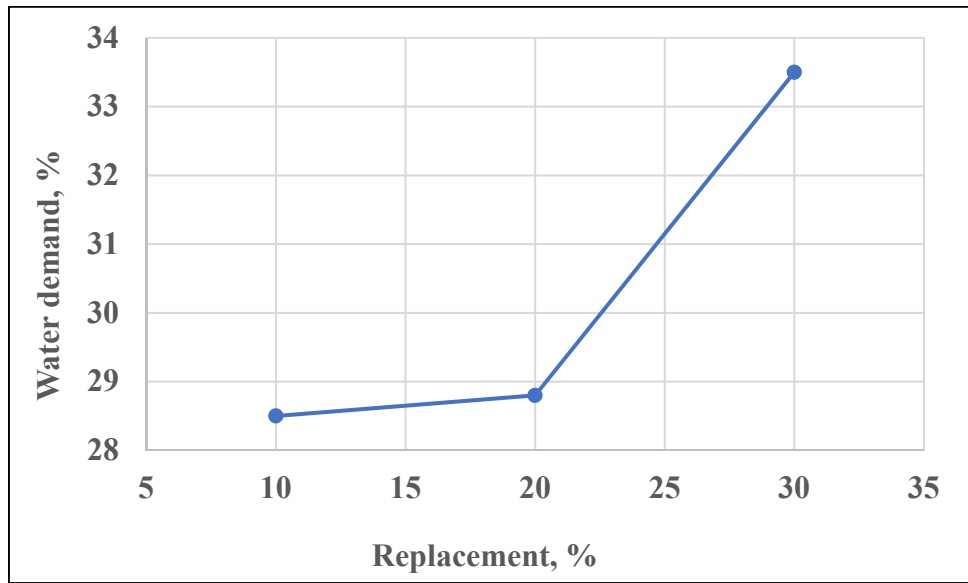

**Figure 8.** Water demand of calcined clay composite cement at different percentage replacements.

The three, seven and 28 days compressive strength values of the CEM-I and composite cements are shown in Figure 9. The control sample obtained a three and seven days' compressive strength of 30 MPa and 36.6 MPa, respectively. These strength values consistently decreased as the calcined clay replacement increased. The addition of calcined clay decreased the three days' strength by 10%, 19% and 48%, respectively, across all replacement levels. Similarly, the 28 days' strength trailed the control by 5.1%, 12.3% and 21.7%, respectively. However, it was observed that the percentage reductions at 28 days were lower than those at three days. This indicates that after 90 days or 180 days, the compressive strength may not follow the same trend. Considering early and ultimate strengths, 20% could be selected as the optimum replacement.

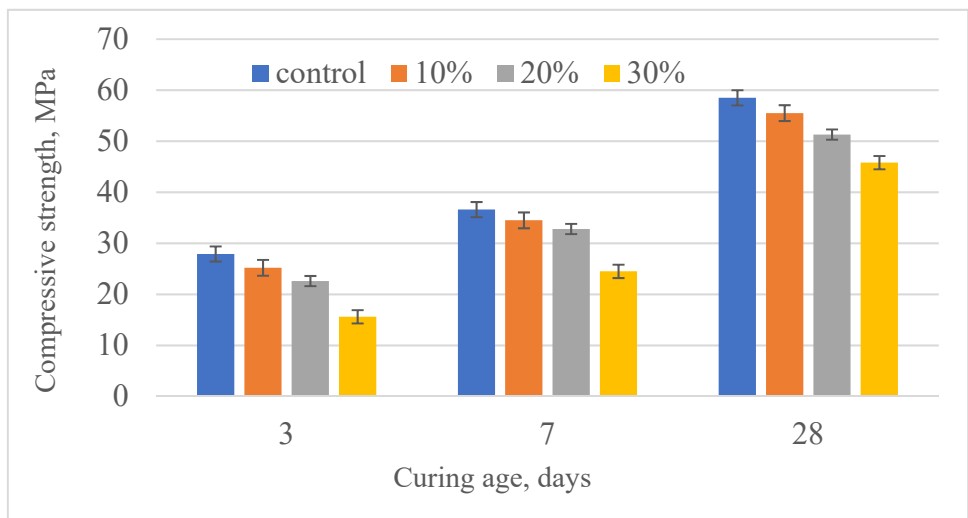

**Figure 9.** Compressive strength results of calcined clay blended mortar at varying percentages.

## 4. Conclusions

This study has shown that low-grade kaolinitic clay has a good potential for being utilised in cementitious systems for construction applications. The chemical make-ups of the raw and calcined clay contained all the relevant oxides and were within the acceptable limits of ASTM C618. The temperature and duration of calcination is very critical, since it affects the phase changes in the material and, consequently, the reactivity of the pozzolan. At a calcination temperature of 800 °C, there was a complete dehydroxylation of kaolinite to form amorphous metakaolinite. The SEM image revealed the transformation of the typical irregular pseudo-laminar structure of the clay into a flaky-like appearance after calcination. The XRD analysis showed that the clay was largely made up of quartz, kaolinite, illite, montmorillonite and other associated minerals.

The Frattini test proved the pozzolanic potential of the calcined clay, as a plot of its CaO and OH$^-$ was found below the Ca(OH)$_2$ solubility curve. More water was required to form a workable paste, consequently increasing the setting time of the composite cement. The incorporation of the calcined clay resulted in decreased compressive strengths at all percentage replacements and curing ages. The optimum replacement between the three levels used was found to be 20 wt.%. Further investigation should be conducted into the hydration and reactivity of calcined low-grade kaolinitic clay and into its long-term durability.

**Author Contributions:** Conceptualization, M.K., E.G., M.S. and A.E.; methodology, M.K., A.E. and K.B.; software, K.B.; validation, A.E., A.D. and M.K.; formal analysis, K.B., M.K., A.E. and E.G.; investigation, K.B. and M.K., resources, A.D., A.E. and K.B.; writing—original draft preparation, K.B.; writing—review and editing, A.D., M.K.; and K.B.; visualization, M.T., E.G. and M.S.; supervision, M.K., M.S., E.G., A.E., A.D. and M.T.; project administration, M.K.; funding acquisition, M.T., A.D. and E.G. All authors have read and agreed to the published version of the manuscript.

**Funding:** This research was funded by Coventry University under project code 17172-01 and the APC was funded by MDPI open access publishing in Basel/Switzerland.

**Institutional Review Board Statement:** Not applicable.

**Informed Consent Statement:** Not applicable.

**Data Availability Statement:** Not applicable.

**Conflicts of Interest:** The authors declare no conflict of interest.

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
