# Peer review of "Mechanochemical Characterisation of Calcined Impure Kaolinitic Clay as a Composite Binder in Cementitious Mortars"

_jcs, doi:10.3390/jcs6050134_

Round 1

Reviewer 1 Report

This manuscript reports the use of low-grade calcined clay as supplementary cementitious materials. Overall, the idea is not so innovative, but the results being provided can be valuable. Below are my comments. The authors need to make improvements on the manuscript before I can re-evaluate the appropriateness for publication.

The introduction part can be improved. The effect of SCMs on the properties of cementitious materials, e.g., hydration and microstructure, and the environmental benefits of SCMs need to be addressed. Here are three valuable papers that need to be included: C Li, et al., Cement and Concrete Research 92 (2017) 98–109; J Li, et al., Resources, Conservation & Recycling 178 (2022) 106038; J Li, et al. Journal of Cleaner Production 261 (2020) 121224.

Table 1. Please clarify what do CC represent in Table 1. Does CC mean calcined clay? If yes, can you explain why the chemical composition of CC are so different from that of raw clay? Also, can you explain why the sum of all the oxides of raw clay is much smaller than 100? Probably, you may need to measure the loss of ignition of the raw materials.

The quality of Figure 2 needs to be improved.

The weight loss below 100 C on the TG curve can be also induced by the loss of free water. I do not think this phenomenon fully supports the presence of montmorillonite.

Figure 3. I think you also need to present the SEM image of raw clay in order to support you discussion.

The XRD patterns need to be double checked. I do not think the peaks have been well assigned.

Author Response

The authors wish to thank the reviewers for their insightful and valuable comments. All corrections and suggestions have been implemented and explained as follows:

Reviewer no. 1:

  1. The introduction part can be improved. The effect of SCMs on the properties of cementitious materials, e.g., hydration and microstructure, and the environmental benefits of SCMs need to be addressed. Here are three valuable papers that need to be included: C Li, et al., Cement and Concrete Research 92 (2017) 98–109; J Li, et al., Resources, Conservation & Recycling 178 (2022) 106038; J Li, et al. Journal of Cleaner Production 261 (2020) 121224.

A few lines addressing the effect of SCMs on the properties of cementitious materials have been added to the introduction. The three valuable papers suggested have also been considered and cited appropriately.

  1. Table 1. Please clarify what do CC represent in Table 1. Does CC mean calcined clay? If yes, can you explain why the chemical composition of CC are so different from that of raw clay? Also, can you explain why the sum of all the oxides of raw clay is much smaller than 100? Probably, you may need to measure the loss of ignition of the raw materials.

‘CC’ in Table 1 has been replaced with ‘calcined clay’. Some elements have been excluded from the table in order to make it more concise. This also includes loss of ignition. The sum of all will bring the percentage to 100. Again, heat treatment of clay at higher temperatures could cause the dissipation of some elements and transform some phases present.

  1. The quality of Figure 2 needs to be improved.

The figure has been replaced with an improved version.

  1. The weight loss below 100 C on the TG curve can also be induced by the loss of free water. I do not think this phenomenon fully supports the presence of montmorillonite.

The authors agree with this comment. The statement has been revised in the text.

  1. Figure 3. I think you also need to present the SEM image of raw clay in order to support you discussion.

The SEM image of raw clay has been included.

  1. The XRD patterns need to be double checked. I do not think the peaks have been well assigned.

The XRD pattern has been checked again, as suggested and the peaks appear to be assigned appropriately. However, reviewer’s thoughts on the peak assignment are gladly welcome for consideration.

Reviewer 2 Report

This report investigated the feasibility of replacing Portland cement with low-grade kaolinitic clays in cement mortar. The physical, chemical, mineralogical and mechanical characteristics of the blended calcined clay cement were determined. The topic is of potential interest for readers. I have several comments for authors' consideration:

  • The English of this paper should be improved. For example, Page 1, it should be “the production of Portland cement” instead of “the production Portland cement”; and Page 3, it should be “reactivity of pozzolans can significantly be affected by how the particles are distributed”. There are some other mistakes in the paper. Please carefully check it.
  • There should be full names for the abbreviations in their first appearances, for example, Supplementary Cementitious Materials (SCMs).
  • Page 2, the fineness of the calcined clay was 21.3 m2/g. Is this number correct? Because it’s much bigger than the fineness of cement.
  • Could the author provide the strength of blended mortar at 90 days? Based on the result shown in this report, it is not convincible to say that “compressive strength is likely to improve and possibly obtain similar or higher values if curing age is increased to 90 or 180 days”.

Author Response

The authors wish to thank the reviewers for their insightful and valuable comments. All corrections and suggestions have been implemented and explained as follows:

Reviewer no. 2

  1. The English of this paper should be improved. For example, Page 1, it should be “the production of Portland cement” instead of “the production Portland cement”; and Page 3, it should be “reactivity of pozzolans can significantly be affected by how the particles are distributed”. There are some other mistakes in the paper. Please carefully check it.

The errors identified by the reviewer has been corrected. The authors have read through again and similar errors have also been corrected.

  1. There should be full names for the abbreviations in their first appearances, for example, Supplementary Cementitious Materials (SCMs).

The full name of SCM has been fully indicated in its first appearance.

  1. Page 2, the fineness of the calcined clay was 21.3 m2/g. Is this number correct? Because it’s much bigger than the fineness of cement.

The calcined clay used for this work is finer than the cement. This was done to increase its surface area, thereby improving reactivity of the calcined clay.

  1. Could the author provide the strength of blended mortar at 90 days? Based on the result shown in this report, it is not convincible to say that “compressive strength is likely to improve and possibly obtain similar or higher values if curing age is increased to 90 or 180 days”.

The research is still ongoing and the 90 days results are not ready. The authors will be happy to share the 90 days results in the next paper. However, the statement has been revised in the manuscript to address the reviewers concerns.

Reviewer 3 Report

This an interesting study about the use of calcinated kaolinitic clay as cement substitute.

The manuscrit is well written.

Authors must improve the quality of their FIGURES before the final acceptance of this paper

Author Response

The authors wish to thank the reviewers for their insightful and valuable comments. All corrections and suggestions have been implemented and explained as follows:

Reviewer no. 2

This an interesting study about the use of calcinated kaolinitic clay as cement substitute.

The manuscript is well written.

Authors must improve the quality of their FIGURES before the final acceptance of this paper

The high-quality figures have been replaced in the edited version.

Round 2

Reviewer 1 Report

The authors have made improvements on the manuscript. The manuscript can be accepted but the following two comments need to be addressed.

1) Please see Table 1 again. By adding up all the elements of RAW clay, we do not get 100%. I request the authors to check what are the remaining. If the remaining parts belong to chemically bound water, then you need to add an extra column of LOSS on IGNITION. Also, please check again what does the Al2O3 come from when RAW clay converses into CALCINED clay.

2) Please make improvements on the quality of figures. All figures should be clear as Fig. 8.

Author Response

I attached the response.
